# *Ganoderma lucidum* (Curtis) P. Karst. Immunomodulatory Protein Has the Potential to Improve the Prognosis of Breast Cancer Through the Regulation of Key Prognosis-Related Genes

**DOI:** 10.3390/ph17121695

**Published:** 2024-12-16

**Authors:** Zanwen Zuo, Ruihua Wen, Shuang Jing, Xianghui Chen, Ruisang Liu, Jianping Xue, Lei Zhang, Qizhang Li

**Affiliations:** 1Innovative Drug R&D Center, Innovative Drug Research Center, College of Life Sciences, Huaibei Normal University, Huaibei 235000, China; zanwenzuo@foxmail.com (Z.Z.);; 2School of Medicine, Shanghai University, Shanghai 200444, China; 3National “111” Center for Cellular Regulation and Molecular Pharmaceutics, Key Laboratory of Fermentation Engineering (Ministry of Education), Hubei Key Laboratory of Industrial Microbiology, Cooperative Innovation Center of Industrial Fermentation (Ministry of Education & Hubei Province), School of Life Science and Health Engineering, Hubei University of Technology, Wuhan 430068, China; 4Department of Pharmaceutical Botany, School of Pharmacy, Naval Medical University, Shanghai 200433, China; 5Key Laboratory of Plant Secondary Metabolism and Regulation of Zhejiang Province, College of Life Sciences and Medicine, Zhejiang Sci-Tech University, Hangzhou 310018, China

**Keywords:** breast cancer, luminal A, prognostic signature, TMEM63C, *Ganoderma lucidum* immunomodulatory protein, prognosis improvement

## Abstract

**Background/Objectives**: Breast cancer in women is the most commonly diagnosed and most malignant tumor. Although luminal A breast cancer (LumA) has a relatively better prognosis, it still has a persistent pattern of recurrence. *Ganoderma lucidum* (Curtis) P. Karst. is a kind of traditional Chinese medicine and has antitumor effects. In this study, we aimed to identify the genes relevant to prognosis, find novel targets, and investigate the function of the bioactive protein from *G. lucidum*, called FIP-glu, in improving prognosis. **Methods**: Gene expression data and clinical information of LumA breast cancer patients were downloaded from The Cancer Genome Atlas (TCGA) and Gene Expression Omnibus (GEO) databases. Using bioinformatics methods, a predictive risk model was constructed to predict the prognosis for each patient. The cell counting kit-8 (CCK8) and clone formation assays were used to validate gene function. The ability of FIP-glu to regulate RNA levels of risk genes was validated. **Results**: Six risk genes (slit-roundabout GTPase-activating protein 2 (SRGAP2), solute carrier family 35 member 2 (SLC35A2), sequence similarity 114 member A1 (FAM114A1), tumor protein P53-inducible protein 11 (TP53I11), transmembrane protein 63C (TMEM63C), and polymeric immunoglobulin receptor (PIGR)) were identified, and a prognostic model was constructed. The prognosis was worse in the high-risk group and better in the low-risk group. The receiver operating characteristic (ROC) curve confirmed the model’s accuracy. Gene Ontology (GO) and Kyoto Encyclopedia of Genes and Genomes (KEGG) enrichment analyses showed that the differentially expressed genes (DEGs) between the high- and low-risk groups were significantly enriched in the immune responses. TMEM63C could promote tumor viability, growth, and proliferation in vitro. FIP-glu significantly regulated these risk genes, and attenuated the promoting effect of TMEM63C in breast cancer cells. **Conclusions**: SRGAP2, SLC35A2, FAM114A1, TP53I11, TMEM63C, and PIGR were identified as the potential risk genes for predicting the prognosis of patients. TMEM63C could be a potential novel therapeutic target. Moreover, FIP-glu was a promising drug for improving the prognosis of LumA breast cancer.

## 1. Introduction

Breast cancer in women is the most commonly diagnosed type of cancer. In China, breast cancer is the fourth most common fatal disease in women, after lung, colon, and stomach cancers, with 420,000 new cases (9.1%) and 12,000 deaths (3.9%). Clinically, breast cancer is generally classified into four molecular subtypes according to the gene expression profiling of estrogen receptor alpha (ERα), progesterone receptor (PR), and human epidermal growth factor receptor 2 (Her2): Luminal (Lum) A, LumB, HER2-enriched (HER2), and triple-negative breast cancer (TNBC) [1]. Drug treatment, radiotherapy, and surgery are the main treatments for breast cancer. Although LumA has a better prognosis compared to the other breast cancer subtypes, it is still the most dominant, as LumA patients account for the highest percentage of total breast cancer patients [2]. It is of great importance to identify effective treatment strategies. In detecting the development of tumors and curing tumors, the poor therapeutic effect is mainly due to the lack of adequate indicators. Therapeutic progress is achieved by developing new drugs that target dysregulated genes. Unfortunately, drug resistance is a significant challenge in treating cancers [3]. There is a great need to find new markers for better prognosis and treatment.

With the development of sequencing technology, a large amount of gene expression information of the various molecular subtypes of breast cancer is captured and recorded in databases, such as Gene Expression Omnibus (GEO) and The Cancer Genome Atlas (TCGA). Microarray data in the database reflect the general conditions of expression information, targeting tens of thousands of genes. Based on bioinformatics techniques, microarray analysis can acquire the expression information and study differentially expressed genes (DEGs) during the occurrence and development of diseases [4]. Using bioinformatics techniques, more and more researchers have studied DEGs in the development of various cancers such as gastric cancer [5], craniopharyngioma [6], bladder cancer [7], etc. After screening DEGs, some essential genes are identified as potential biomarkers. Using this strategy, several potential biomarkers in TNBC [8,9], HER2 [10,11], and LumA/B [12] have been reported. Some researchers have identified critical DEGs between different breast cancer subtypes [13,14].

*Ganoderma lucidum* (Curtis) P. Karst. has been a well-known traditional medicine for two thousand years and is also considered a new resource of food in China. A fungal immunomodulatory protein from *G. lucidum*, called Lingzhi-8, LZ-8, or FIP-glu [15,16], is a kind of bioactive protein with immunoregulating activity [17], such as the activation of mouse spleen cells and macrophage cells [16,18]. FIP-glu also has antitumor effects and can inhibit the growth of various cancer cells [19]. Its antitumor effect can be achieved through several molecular mechanisms in different ways. Therefore, FIP-glu is a promising application for medicinal use [17].

The present study used the GEO and TCGA databases to identify robust DEGs in LumA breast cancer. A reliable and user-friendly prognostic model was constructed using these DEGs and the corresponding clinical data. The tumor-promoting effect of a potential novel therapeutic target transmembrane protein 63C (TMEM63C) was investigated in vitro. Then, it was found that the *G. lucidum* immunomodulatory protein FIP-glu significantly regulated the mRNA levels of risk genes and reversed the tumor-promoting activity of TMEM63C in breast cancer T-47D and MCF-7 cells. This study provided reliable molecular biomarkers for the screening and diagnosis of LumA breast cancer and a prognostic model that could help clinicians estimate the risk of death in LumA patients. The pro-tumorigenic effect of TMEM63C was first validated in breast cancer cells, and this gene may be a potential novel therapeutic target. In addition, it also provided a molecular basis for that FIP-glu in improving the prognosis of patients with LumA breast cancer.

## 2. Results

### 2.1. Identification and Functional Enrichment Analysis of DEGs

According to the threshold adjusted *p* < 0.05, 7679 and 15,293 DEGs were found in the LumA subtype from GSE45827 and TCGA, respectively (Figure 1A,B). The Venn plot showed that 4564 DEGs were commonly present (Figure 1C,D). Gene Ontology (GO) term functional enrichment and Kyoto Encyclopedia of Genes and Genomes (KEGG) pathway enrichment analyses of these genes were performed to gain further biological understanding. A total of 1090 significant GO terms were associated with these DEGs (adjusted *p* < 0.05). The top ten enriched GO terms for biological processes (BP), cellular components (CC), and molecular function (MF) are shown in Figure 1E. KEGG pathway enrichment analysis revealed that 58 pathways were significantly enriched at an adjusted *p* < 0.05 (Figure 1F), including breast cancer, ErbB, DNA replication, cell cycle, and Ras signaling pathways.

### 2.2. Construction of the Prognostic Model

The mRNA expression information of 413 breast cancer and 113 normal samples and the corresponding clinical information of the patients were downloaded from TCGA. Among the 4564 DEGs, univariate Cox regression analysis showed that 117 genes were significantly correlated with overall survival (OS) using a *p* value < 0.05 as the filtering criterion (Appendix A). Then, last absolute shrinkage and selection operator (LASSO)-Cox regression was applied to shrink predictors (Figure 2A,B) and showed that 34 mRNAs were suitable for building the prognostic risk model. The mRNAs were entered into a multivariate Cox regression analysis to construct a prognostic risk model. Finally, six genes, slit-roundabout GTPase-activating protein 2 (SRGAP2), solute carrier family 35 member 2 (SLC35A2), sequence similarity 114 member A1 (FAM114A1), tumor protein P53-inducible protein 11 (TP53I11), TMEM63C, and polymeric immunoglobulin receptor (PIGR), were used to establish a prognostic model (Figure 2C and Table 1): Risk score = SRGAP2 _Exp_ × 1.3163 + SLC35A2 _Exp_ × 1.0448 + FAM114A1 _Exp_ × 0.9542 + TP53I11 _Exp_ × 0.8025 + TMEM63C _Exp_ × 0.7617 + PIGR _Exp_ × (−0.3600). The heatmap and boxplot diagram showed that all the mRNAs had statistical differences between the normal and tumor samples (Figure 2D and Appendix A). The high-risk score predicted a poor prognosis for patients. The higher the prognostic score, the more the death in patients. Patients could be classified into two groups, low- (*n* = 197) and high-risk (*n* = 216), according to the median risk score (1.1566), which was reflected by principal component analysis (PCA) (Figure 2E,F). In addition, patients with high-risk score had lower survival rate compared with ones with low-risk score (*p* = 4.22 × 10^−6^; Figure 2G and Appendix A). The area under curve (AUC) of the receiver operating characteristic (ROC) curve was used to assess the prognostic ability of the model. The AUC value reached 0.750 at one year, indicating that the model showed high sensitivity and specificity (Figure 2H and Appendix A).

### 2.3. Independent Prognostic Value and Clinical Performance of the Risk Score

The scatter plot shows that the patients with lower risk scores had more prolonged survival (Figure 3A and Appendix A). The heatmap shows that the risk genes significantly differed between the low- and high-risk groups. The 1-, 3-, and 5-year AUCs were 0.745, 0.835, and 0.777, respectively (Figure 3B). Univariate Cox regression analysis showed that risk score was significantly related to prognosis, and the score could be used as an independent prognostic factor based on multivariate risk regression analysis (Figure 3C). The AUCs at 1-, 3-, and 5-year for risk score and age were 0.750 and 0.933, 0.857 and 0.805, and 0.784 and 0.792 (Figure 3D).

As shown in Figure 3E, model gene expressions were significantly different between the high- and low- risk groups (*p* < 0.05). The heatmap shows a significant age difference (≤60 or >60 years; *p* = 0.035). Some clinical factors and some genes also showed statistical differences (Appendix A). To further verify the effectiveness of the risk model in predicting prognosis, patients were divided into high- and low-risk groups by age (≤60 and >60 years), stage (stage I + II and stage III + IV), N stage (N0, N1, and N2), and T stage (T1 + 2 and T3 + 4) based on the median risk score. The Kaplan–Meier (KM) survival curve showed that the patients with high-risk scores had shorter OS in all subgroups (Figure 3F).

To provide a clinically available and practical tool for predicting the overall survival of LumA patients, two nomograms were constructed to reveal the 1-, 5-, and 9-year survival rates. The one combined risk score with clinical factors such as age, stage, N stage, and T stage (Appendix A). The C-index of the nomogram was 0.8363 ± 0.0355, indicating that this nomogram had exceptional stability. A calibration curve was used to compare the consistency of actual and predicted 1-, 5-, and 9-year patient survival. The results showed that the actual line matched well compared to the predicted line (Appendix A). In addition, the other nomogram was constructed to examine the association between overall survival and the expression of model genes, including SRGAP2, SLC35A2, FAM114A1, TP53I11, TMEM63C, and PIGR (Appendix A). Its C-index was 0.7804 ± 0.0356, and the calibration plots suggest a well-fitted nomogram (Appendix A).

### 2.4. Functional Enrichment Analysis of DEGs Between High- and Low-Risk Score Groups

GO and KEGG enrichment analyses were performed to investigate the biological function of DEGs between high- and low-risk score groups and the risk score of model genes. Using the R package limma, 1586 DEGs were obtained with adjusted *p* < 0.05 and |logFC| > 0.5. GO enrichment analysis showed that these DEGs were enriched in various immune-related terms, such as T-cell activation, regulation of T-cell activation, lymphocyte migration, mononuclear cell differentiation, cytokine activity, chemokine activity, and immune receptor activity (Figure 4A). Similar observations were found in KEGG enrichment analysis (Figure 4B). Gene Set Enrichment Analysis (GSEA) was then carried out and the relative pathways are shown in Figure 4C.

### 2.5. Consensus Clustering Analysis of the Model Genes

Consensus clustering was conducted on the LumA samples based on the expression of genes in the prognostic risk model to explore the different features of function. According to the cumulative distribution function curve (Figure 5A) and the delta area curve (Figure 5B), k = 2 was the optimal number and the samples were divided into two subtypes, C1 (*n* = 174) and C2 (*n* = 239) (Figure 5C). Between the two subgroups, 518 DEGs were identified, including 425 up- and 93 down-regulated genes (adjusted *p* < 0.05 and |logFC| > 0.5). Through GO analysis, the DEGs were related to regulation of T-cell activation, regulation of T-cell proliferation, regulation of lymphocyte proliferation, regulation of mononuclear cell proliferation, immune receptor activity, MHC protein binding, and Toll-like receptor binding (Figure 5D). KEGG pathway analysis revealed that these DEGs were associated with immune-related pathways (Figure 5E). GSEA then showed endocytosis and type II diabetes mellitus pathways were up-regulated in C1, while cytokine-cytokine receptor interaction, autoimmune thyroid disease, intestinal immune network for IgA production, etc. were up-regulated in C2 (Figure 5F).

### 2.6. FIP-Glu-Regulated Model Genes in Breast Cancer Cells

The expression of six potential mRNAs was examined in the breast cancer cell lines T-47D and MCF-7 with or without FIP-glu treatment using RT-qPCR. As shown in Figure 6A, FIP-glu could suppress the transcriptional levels of FAM114A1, SLC35A2, SRGAP2, TMEM63C, and TP53I11, and enhanced the PIGR mRNA level in T-47D cells. Similar results were observed in MCF-7 cells (Figure 6B); however, FAM114A1 and PIGR were not detected due to the lower transcriptional levels.

### 2.7. TMEM63C Promotes Tumor Growth in Breast Cancer Cells

The effect of TMEM63C on breast cancer cells has not been reported. To determine the role of TMEM63C in the malignant potential of breast cancer cells, we overexpressed this gene in the T-47D and MCF-7 cells (Appendix A). The Cell Counting Kit (CCK-8) assay revealed that the breast cancer cell viability was significantly increased after TMEM63C overexpression compared to the control group (*p* < 0.0001; Figure 7A). The clone formation assay revealed that the clonogenic ability of T-47D cells overexpressing TMEM63C was significantly increased compared to that of control cells (*p* < 0.0001; Figure 7B); similar results were observed with the MCF-7 cell line (Figure 7). Based on these results, we can speculate that TMEM63C promotes the growth and proliferation of breast cancer cells.

### 2.8. FIP-Glu Attenuates the Pro-Oncogenic Effect of TMEM63C on Breast Cancer Cells

To verify the antitumor effects of FIP-glu through regulating this critical gene, we used FIP-glu to treat TMEM63C-overexpressing T-47D and MCF-7 cells. The CCK-8 results showed that FIP-glu significantly reduced the promoting effect of TMEM63C on cell viability in T-47D and MCF-7 cells (*p* < 0.0001; Figure 8A). The clone formation assay also revealed that FIP-glu attenuated the enhanced effect of TMEM63C on clonal formation ability (*p* < 0.0001; Figure 8B).

## 3. Discussion

Female breast cancer is the most commonly diagnosed malignancy with a high mortality rate among women worldwide. More than 70% of breast cancer cases are ERα+, such as the LumA subtype. Despite the relatively good prognosis of patients with the luminal breast cancer subtype [20], one study showed that the median 15-year relapse rate is 27.8% for LumA [21]. Combining earlier diagnosis and improved detection is a more effective regimen for improving overall survival. Therefore, identifying more reliable biomarkers and targets for breast cancer diagnosis and treatment is necessary.

Using the GEO and TCGA database, many researchers screened essential genes for triple-negative [8], HER2-positive [11], and other types of breast cancer [22]. Key genes were also identified between different breast cancer subtypes. For example, Wang et al. screened immunosuppressive factors in luminal- and basal-like BC cell lines and tissue samples [12], and Xu et al. explored the genetics commonness between basal-like and luminal subtypes of muscle-invasive bladder cancer and breast cancer [23]. Based on the analysis of four breast cancer subtypes, Kim et al. determined the differentially expressed proteins related to lipid metabolism [24], while Malvia et al. investigated the pathways and genes associated with breast tumorigenesis in Indian women [25]. Similarly, six critical genes (SRGAP2, SLC35A2, FAM114A1, TP53I11, TMEM63C, and PIGR) were identified in the present study. Based on the prognostic model constructed by the genes, the low-risk group had higher survival rates. FAM114A1, the family with sequence similarity 114 member A1, has been associated with several diseases such as heart disease [26] and cancers [27,28]. In BC, FAM114A1 was identified as a risk gene [29]. SLC35A2, which encodes the uridine diphosphate galactose, is a solute carrier 35 (SLC35) family member. Sun et al. showed that SLC35A2 had good diagnostic and prognostic values in several cancer types [30]. They also showed that SLC35A2 expression level was increased in breast cancer and knockdown of SLC35A2 could inhibit tumor growth in vivo. The physiological roles of SRGAP2, the slit-roundabout GTPase-activating protein 2, have been extensively studied in the function of cortical neurons [31]. It has also been identified as a potential oncogene and prognostic biomarker in hepatocellular carcinoma [32]. Wang et al. found that reduction of SRGAP2 targeted by exosomal miR-29b secreted from pancreatic cancer cells inhibits angiogenesis by human umbilical vein endothelial cells [33]. SRGAP2 was involved in cell membrane outward bending and regulated neuronal migration and morphogenesis [34]. SRGAP2 was also involved in breast cancer cell migration and the cancer process [35]. Transmembrane protein (TMEM)-family proteins such as TMEM63A, TMEM63B, and TMEM63C were the mammalian orthologs of AtCSC1 and OSCA1, which were capable of forming hyperosmolarity activated cation channels [36]. In human podocytes, down-regulated TMEM63C decreased cell viability and increased cell apoptosis, suggesting that TMEM63C was a potential pro-survival factor [37]. TMEM63C has been implicated in many clinical diseases, such as Alzheimer’s disease [38]. In cancer, a high expression level of TMEM63C was associated with a worse prognosis in patients with papillary thyroid carcinoma [39], and TMEM63C was negatively associated with high lymph node proportion in pancreatic cancer [40]. The functions of TMEM63C in breast cancer have not been reported. Based on ENCORI, TMEM63C RNA level was increased in breast cancer (Appendix A). Thus, TMEM63C was overexpressed in T-47D and MCF-7 cells, and it was found that the viability and clone formation ability were significantly improved, suggesting that TMEM63C may be a potential target in breast cancer treatment. TP53I11, the tumor protein P53-inducible protein 11, was a transcriptional target of TP53 [41]. A pan-cancer analysis showed that high TP53I11 expression was significantly associated with poor overall survival in breast cancer [42]. Xiao et al. found that TP53I11 can suppress epithelial–mesenchymal transition and metastasis in breast cancer cells [43]. Similarly, overexpressed TP53I11 induced apoptosis in human hepatocellular carcinoma HepG2 cells [44]. Therefore, a more in-depth investigation is required. PIGR, polymeric immunoglobulin receptor, is responsible for the capture and transcytosis of dimeric IgA (dIgA) and plays a vital role in mucosal immunity [45]. In addition, studies have shown that PIGR is involved in cancers and has prognostic values in pancreatic cancer [46], ovarian cancer [47], adenocarcinoma of the upper gastrointestinal tract [48], etc. The role of PIGR in cell proliferation varies among cancer cell types [49,50]. In breast cancer, PIGR is shown to be downregulated in tumor tissues compared to normal tissue [51], and high PIGR expression is strongly associated with better cumulative survival compared to low PIGR levels [52].

The immune system is central to tumor development and progression [53]. Evidence shows that modulation of the immune system represents promising therapeutic strategies in cancer therapy [54]. Through GO and KEGG enrichment analyses of the DEGs between high- and low-risk groups, we found that these DEGs are mainly involved in the immune response. It can be hypothesized that immune level influences tumor progression and the prognosis of patients. Consensus clustering is a widely used approach to explore subclasses using gene expression data. Wei et al. identified three distinct clusters based on the expression similarity of 139 redox genes in TCGA that had significant prognostic value and clinical association [55]. Using consensus clustering analysis, Li et al. divided the patients into two clusters with different survival prognoses [56]. Based on the expression of genes in the prognostic risk model, we performed consensus clustering analysis and identified two clusters. Surprisingly, the functions of the DEGs between clusters are still involved in immune functions. Therefore, immunotherapy would be a better intervention in the treatment of LumA breast cancer.

*G. lucidum* is a medicinal mushroom widely used in China for thousands of years. Numerous pharmacologically active compounds have been shown to contribute to the antitumor activity of *G*. *lucidum* in many studies. Among these compounds, in addition to polysaccharides and triterpenoids, the immunomodulatory protein (FIP-glu) is an essential bioactive protein with various biological functions, such as immunomodulation and antitumor effects [16,19,57]. FIP-glu can inhibit a variety of tumors; for example, it promotes apoptosis with the enhancement of caspase-3 activity in human leukemia cells [58], inhibits the growth of lung cancer cells via p53-dependent G1 arrest [59], and suppresses the viability of mouse melanoma cells [60]. However, the effect of FIP-glu on inhibiting breast cancer cells has never been reported. An immunomodulatory protein from *G. atrum*, discovered by J.D. Zhao, L.W. Hsu, and X.Q. Zhang (FIP-gat) and a homolog of FIP-glu, inhibited cell growth, triggered cell cycle arrest, and induced apoptosis in breast cancer MDA-MB-231 cells [61]. It was hypothesized that FIP-glu would have similar effects on breast cancer cells due to its high homology [62]. This result, of course, was expected (Figure 8A,B). Therefore, in the present study, the prediction model was constructed. Identifying key gene is of critical importance for developing targeted agent [63]. Either directly or indirectly targeting a druggable gene is an effective means of treatment [64,65]. Six prognosis-related key genes including five up- (FAM114A1, SLC35A2, SRGAP2, TMEM63C, and TP53I11) and one down-regulated gene (PIGR) were identified. Unexpectedly, FIP-glu regulated and reversed all these gene expression levels. In the TMEM63C-overexpressing T-47D and MCF-7 cells, FIP-glu still exhibited antitumor effect. The results of cell viability and clone formation assays also indicated that TMEM63C overexpression partly recovered tumor growth in T-47D and MCF-7 cells. These findings reveal a novel regulatory mechanism by which FIP-glu exhibited effects of antitumor and prognosis improvement.

## 4. Materials and Methods

### 4.1. Reagents

Recombinant FIP-glu was prepared from *Pichia pastoris* and stored in our laboratory [62]. The CCK-8 was purchased from Yeasen Biotechnology (Shanghai, China).

### 4.2. Data Source and Preprocessing

The GSE45827 dataset was downloaded from the GEO database (https://www.ncbi.nlm.nih.gov/geo/, public on 24 March 2016, accessed on 1 April 2022). A total of 29 LumA breast tumor samples and 11 normal breast samples were used for analysis. The RNA-seq data and corresponding clinical information were downloaded from TCGA (https://www.cancer.gov/about-nci/organization/ccg/research/structural-genomics/tcga, accessed on 1 April 2022). Patients with a survival time of less than 30 days and missing required data were excluded. A total of 1226 clinical information items were obtained, with 413 LumA breast tumor samples and 113 normal samples collected for analysis. The expression files were analyzed using R (version 3.6.3). The R package limma (version 3.42.0) was used to identify DEGs (differentially expressed genes). An adjusted *p* < 0.05 is considered statistically significant.

### 4.3. Construction of the Prognostic Model

Based on the breast cancer samples and clinical information in the TCGA dataset, the DEGs were subjected to an investigation of the associations between mRNA expression levels and OS with *p* < 0.05 by univariate Cox regression analysis using the R package survival (version 3.1-12). The R package glmnet (version 4.1-4) was used to perform LASSO-Cox regression analysis for selecting the most useful prognostic markers further. Multivariate Cox regression analysis was used to screen independent prognostic DEGs with *p* < 0.05 as the threshold of significant correlation and construct a prognostic model. Breast cancer samples were randomly divided into two groups, the training and test sets, with a 1:1 ratio. The training set was used to construct a prognostic model, while the test and entire sets were used to validate the model. The results of the entire set were also displayed. Based on the expression levels of these genes and coefficients (coefs) from the multivariate Cox proportional hazards regression analysis, a prognostic risk score formula was defined: Risk score = $\sum_{i=0}^{n}gene\;expression\;level\times coef$. Then, based on the median value of risk score, patients with breast cancer were divided into low- and high-risk groups. PCA analysis was used to examine the distribution of the low- and high-risk groups. The R package survival (version 3.1-12) was used to construct the KM curve for the assessment of the prognostic value of the risk score. Using the R package survivalROC (version 1.0.3), ROC curve analysis compared sensitivity and specificity. Several traditional clinical risk factors such as age, stage, T stage, N stage, and risk score were compared using the univariate and multivariate Cox analysis. The nomogram was constructed based on the multivariate Cox analysis using the R package rms (version 6.1-1).

### 4.4. Functional Enrichment Analysis

GO functional enrichment analysis and KEGG pathway enrichment analysis were performed using the R packages GO.db (version 3.10.0) and org.Hs.eg.db (version 3.10.0). GSEA was performed using GSEA software (version 4.2.3) to analyze the pathways in which the DEGs were enriched. *p* < 0.05 was considered to indicate statistical significance. The R package GSVA (version 1.40.1) was used to calculate the infiltration scores of immune cells and the activities of immune-related pathways.

### 4.5. Cell Culture

Breast cancer T-47D and MCF-7 cells were obtained from the China Center for Type Culture Collection (Wuhan, China). The cells were cultured as previously described in [62].

### 4.6. RNA Extraction, cDNA Synthesis, and RT-qPCR

The total RNA extraction (MiniBEST Universal RNA Extraction Kit; Takara, Beijing, China), cDNA synthesis (PrimeScriptTM RT Master Mix (Perfect Real Time); TaKaRa, Beijing, China), and RT-qPCR (SYBR qPCR Master Mix; Vazyme, Nanjing, China) were performed according to the manufacturer’s protocols and as previously described [18]. The primers used for RT-qPCR in this study are listed in Table 2.

### 4.7. Cell Transfection

The *TMEM63C* gene was cloned from cDNA, and subcloned into the p3×FLAG-CMV-10 expression vector to obtain the recombinant plasmid p3FLAG-TMEM63C. Lipofectamine 2000 reagent (Invitrogen, Carlsbad, CA, USA) was used to transfect cells with p3FLAG-TMEM63C or an empty p3×FLAG-CMV-10 vector as a negative control.

### 4.8. Proliferation Assay

Cell viability [66] and plate clone formation assays [67] were performed as previously described.

### 4.9. Statistical Analysis

Cell-related experiments were repeated at least three times. R (Version 4.1.1) and Prism 9 (Version 9.0.0) were used to create graphs. Statistical analyses were performed in R and Prism 9. Data are expressed as mean ± standard deviation (SD).

## Figures and Tables

**Figure 1 pharmaceuticals-17-01695-f001:**
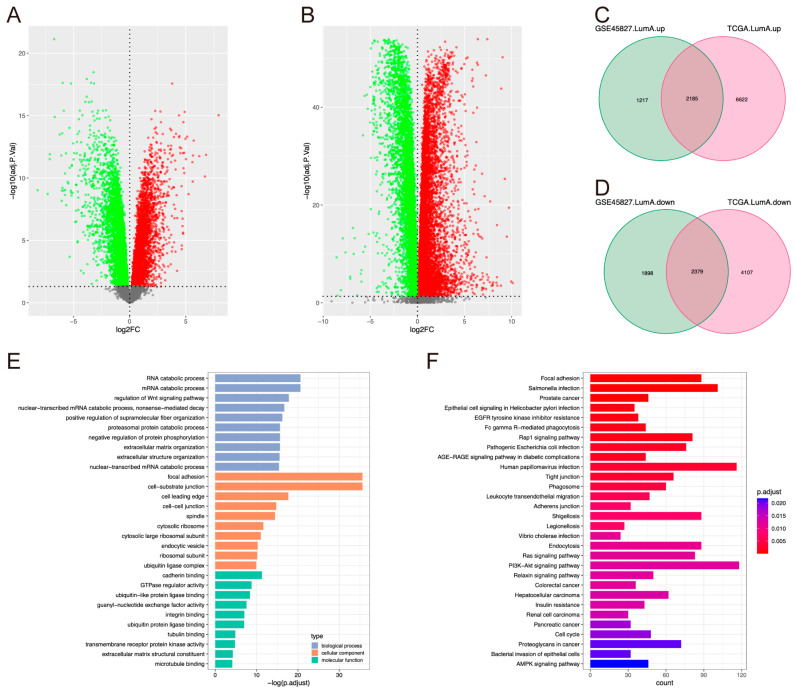
Identification and functional enrichment analysis of DEGs. (**A**,**B**) The volcano plot of DEGs in the Luminal A (LumA) subtype from the GSE45827 (**A**) and TCGA (**B**) datasets. The red points represent up-regulated genes, the green points represent down-regulated genes, and the gray points represent non-significantly regulated genes. The cutoff criterion is adjusted *p* < 0.05. (**C**,**D**) The Venn diagram of up-regulated (**C**) and down-regulated (**D**) DEGs of the Luminal A (LumA) subtype from the GSE45827 and TCGA datasets. (**E**) The bar plot of the GO functional enrichment analysis. The top 10 terms of BP, CC and MF are shown. BP, biological processes; CC, cellular components; and MF, molecular function. (**F**) The bar plot of the KEGG functional enrichment analysis. The top 30 terms are shown.

**Figure 2 pharmaceuticals-17-01695-f002:**
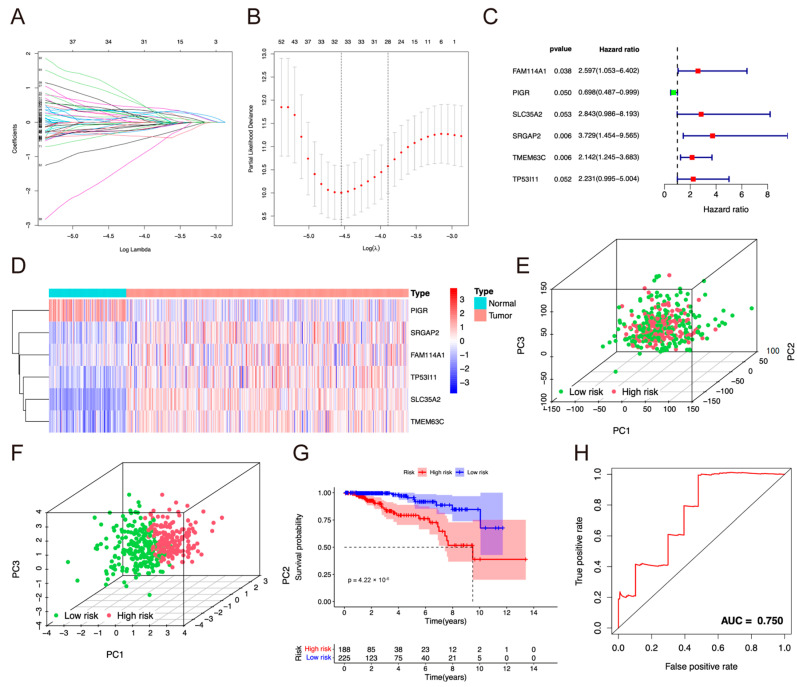
Analysis of the prognostic model. (**A**) LASSO regression of the 117 screened mRNAs. (**B**) Cross-validation for optimizing the parameter in the LASSO regression. (**C**) Forest plots to show the results of the multivariate Cox regression analysis between gene expression and OS. (**D**) Heatmap diagram of the model-related mRNAs in normal and tumor tissues from TCGA. (**E**) PCA analysis of the LumA BC samples from TCGA. (**F**) PCA analysis indicated that the patients possessed significantly high- or low-risk distribution patterns. (**G**) The KM survival analysis of OS between the high-risk group and low-risk group. (**H**) AUC in the ROC analysis for risk signature at 1-year survival time.

**Figure 3 pharmaceuticals-17-01695-f003:**
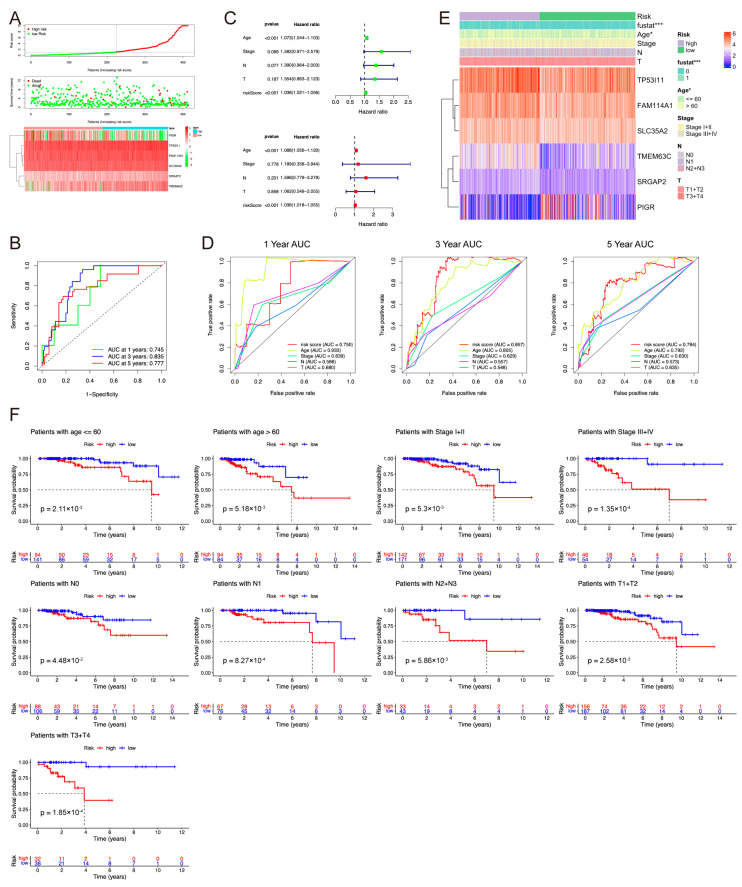
Construction of the prognostic risk model. (**A**) Distribution of the risk score and the expression of risk genes in the entire TCGA set. (**B**) Time-dependent ROC curve analysis of the risk score model for predicting 1-, 3-, and 5-year OS in the entire TCGA set. (**C**) Univariate (green) and multivariate (red) Cox regression analyses for prognostic risk model and clinicopathological characteristics. (**D**) Time-dependent ROC curve analyses of the prognostic variables in the entire TCGA set at five years. (**E**) Heatmap displaying the difference of clinical characteristics between the high- and low-risk groups. *, *p* < 0.05; and ***, *p* < 0.001. (**F**) Survival analysis on clinical features including age, TNM stage, N stage, and T stage.

**Figure 4 pharmaceuticals-17-01695-f004:**
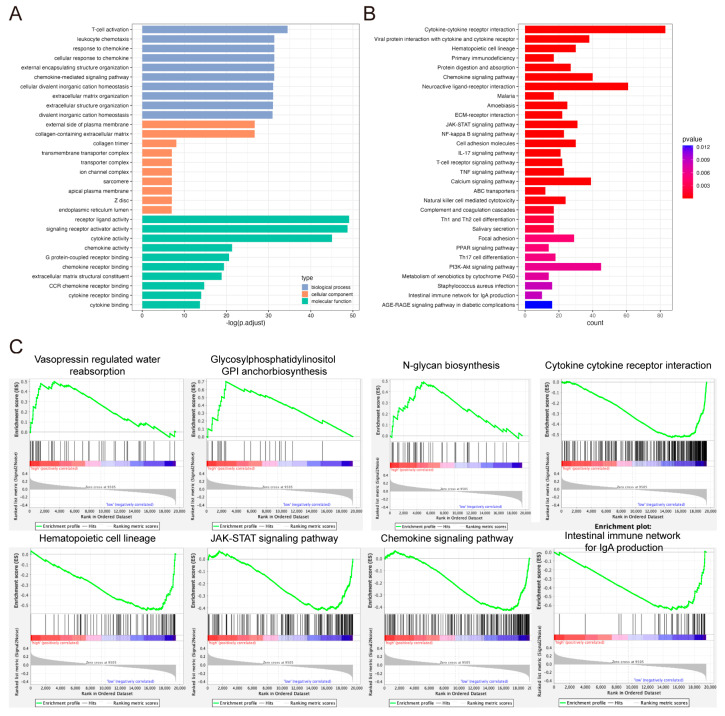
Functional enrichment analysis of DEGs between high- and low-risk score groups. (**A**) The bar plot of the GO functional enrichment analysis. (**B**) The bar plot of the KEGG functional enrichment analysis. (**C**) The GSEA analysis.

**Figure 5 pharmaceuticals-17-01695-f005:**
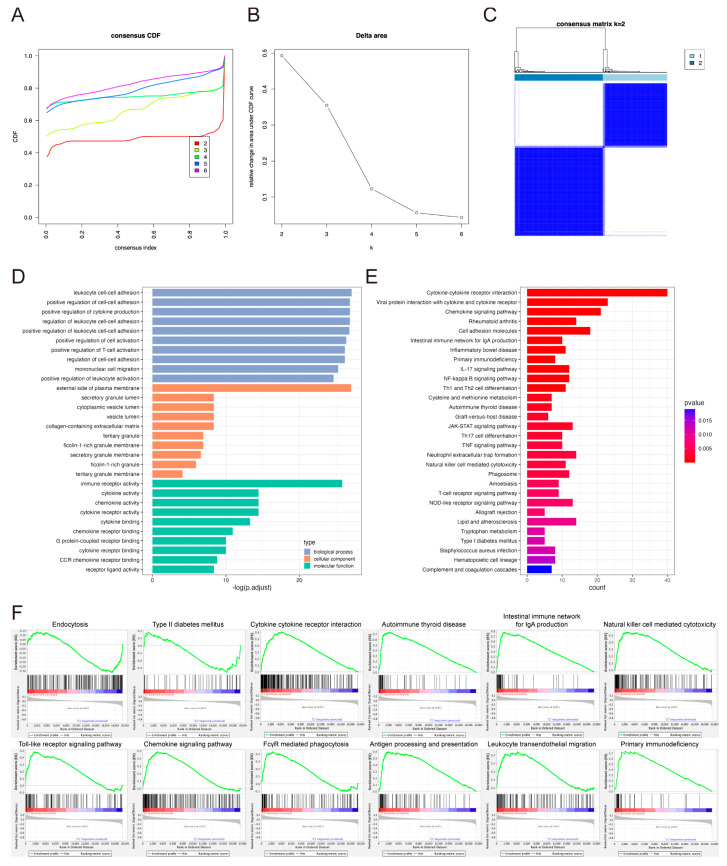
Consensus clustering analysis of expression pattern of genes in the prognostic risk model. (**A**) Cumulative distribution function (CDF) of consensus clustering by consistency analysis when k  =  2–6. (**B**) Relative change of the AUC of the CDF curve when k  =  2–6. (**C**) Consensus matrix heatmap for k = 2. (**D**) The bar plot of the GO functional enrichment analysis. (**E**) The bar plot of the KEGG functional enrichment analysis. (**F**) The GSEA analysis.

**Figure 6 pharmaceuticals-17-01695-f006:**
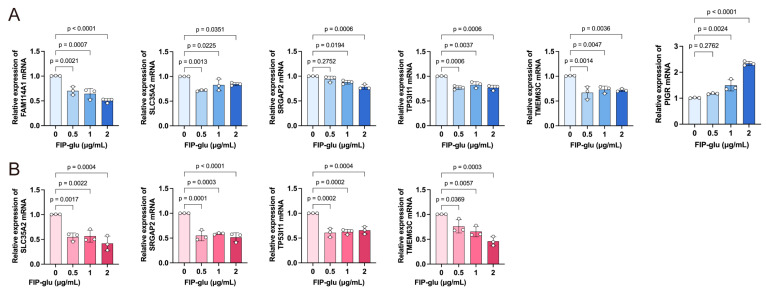
RNA levels of risk genes SRGAP2, SLC35A2, FAM114A1, TP53I11, TMEM63C, and PIGR regulated by FIP-glu in breast cancer T-47D (**A**) and MCF-7 (**B**) cells.

**Figure 7 pharmaceuticals-17-01695-f007:**
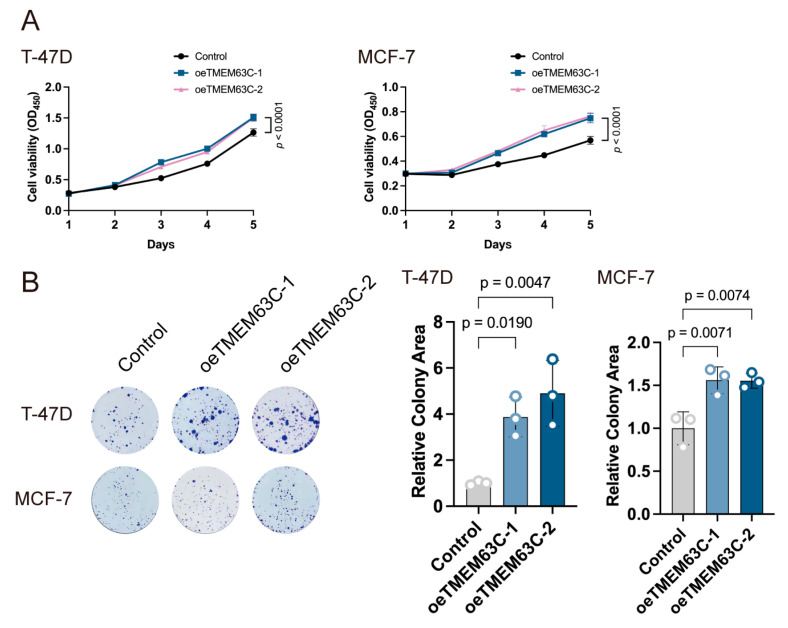
Effect of TMEM63C overexpression on cell growth and proliferation. (**A**) CCK-8 assay revealed the enhanced viability by TMEM63C overexpression in both breast cancer T-47D and MCF-7 cells. Data are presented as mean ± SD (*n* = 6). (**B**) Clone formation assay showed the increase of clone formation ability by TMEM63C overexpression in both breast cancer T-47D and MCF-7 cells. Data are presented as mean ± SD (*n* = 3).

**Figure 8 pharmaceuticals-17-01695-f008:**
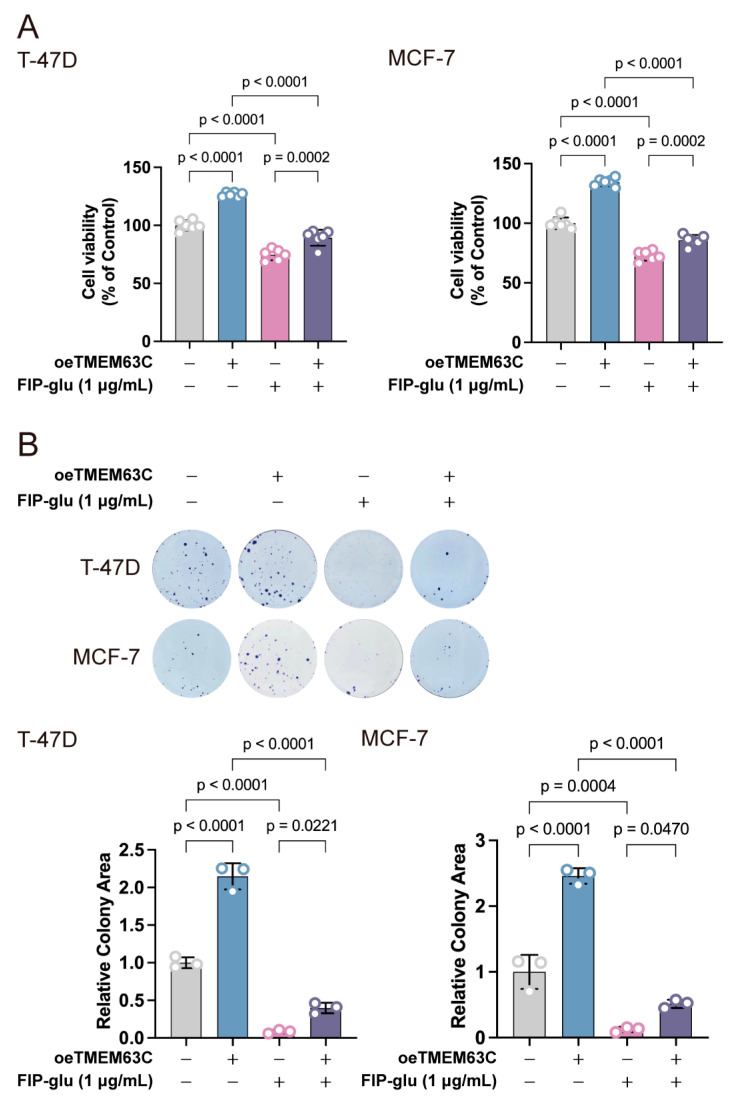
FIP-glu alleviated TMEM63C-mediated pro-proliferative effects. (**A**) CCK-8 assay revealed that FIP-glu inhibited the TMEM63C-induced enhanced viability in T-47D and MCF-7 cells. Data are presented as mean ± SD (*n* = 6). (**B**) Clone formation assay revealed that FIP-glu inhibited the TMEM63C-induced increase of clone formation ability in T-47D and MCF-7 cells. Data are presented as mean ± SD (*n* = 3).

**Table 1 pharmaceuticals-17-01695-t001:** Multivariate Cox analysis.

Id	Coef	HR	HR.95L	HR.95H	*p* Value
SRGAP2	1.3163	3.7294	1.4540	9.5655	0.0062
SLC35A2	1.0448	2.8427	0.9863	8.1931	0.0531
FAM114A1	0.9542	2.5965	1.0531	6.4018	0.0382
TP53I11	0.8025	2.2311	0.9948	5.0041	0.0515
TMEM63C	0.7617	2.1418	1.2454	3.6834	0.0059
PIGR	−0.3600	0.6976	0.4871	0.9993	0.0495

Coef, coefficient; HR, hazard ratio; FAM114A1, sequence similarity 114 member A1; PIGR, polymeric immunoglobulin receptor; SLC35A2, solute carrier family 35 member 2; SRGAP2, slit-roundabout GTPase-activating protein 2; TMEM63C, transmembrane protein 63C; and TP53I11, tumor protein p53-inducible protein 11.

**Table 2 pharmaceuticals-17-01695-t002:** Primers used in this study.

Gene	Direction	Sequence	Purpose
SRGAP2	sense	5′-ccagcaagagacagagcagtt-3′	RT-qPCR
	antisense	5′-aggtcatgcttggcttgtaac-3′	
SLC35A2	sense	5′-catcctcagcatccgctacg-3′	RT-qPCR
	antisense	5′-accaggtgcttcacgttacc-3′	
FAM114A1	sense	5′-atgctggtgacaccttagcc-3′	RT-qPCR
	antisense	5′-gatgaaggggaagtgggtgg-3′	
TP53I11	sense	5′-tcagccaggtcttaggcaatg-3′	RT-qPCR
	antisense	5′-agagcacagcagagacgaac-3′	
TMEM63C	sense	5′-gctagaggagcagctaacgg-3′	RT-qPCR
	antisense	5′-tggagtcctggaaggtgaca-3′	
PIGR	sense	5′-aaaggcggtggcagatacaa-3′	RT-qPCR
	antisense	5′-aactcggtcgacgttcttcc-3′	
GAPDH	sense	5′-ccgcatcttcttttgcgtcg-3′	RT-qPCR
	antisense	5′-gcccaatacgaccaaatccgt-3′	
cTMEM63C	sense	5′-cttgcggccgcagaattcatgtctgcctcaccagacgac-3′	Clone
	antisense	5′-ggtaccgatatcagatcttcagtggtactggttctggcc-3′	

## Data Availability

The original data presented in the study are openly available in TCGA at https://www.cancer.gov/ccg/research/genome-sequencing/tcga (accessed on 1 April 2022) and GEO (GSE45827).

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
