# Peer review of "Ganoderma lucidum (Curtis) P. Karst. Immunomodulatory Protein Has the Potential to Improve the Prognosis of Breast Cancer Through the Regulation of Key Prognosis-Related Genes"

_pharmaceuticals, 2024, doi:10.3390/ph17121695_

Round 1

Reviewer 1 Report

Comments and Suggestions for Authors

This is an important paper. The Authors suggest new, genetically based ways for the early diagnosis of one of the most dangeres variants of breast cancer, based on the use of Gandoderma lucidum, known also from Chinese traditional medicine. The genes relevant to the prognosis of this dangerous variant of cancer became characterized which gives the research a more solid molecular basis. The results can be used in clinical praxis relatively soon.

It is a particular merit of the manuscript that several "very-last" publications are cited from 2023 and even 2024.

Some stylistic defects can be certainly corrected by the Authors. First of all the use of abbreviations should be explained at their first appearance in the manuscript (even in the Abstract!). Secondly a mother language control of the English appears to be desirable, for example already the TITLE is formulated in a manner that one can/could not exactly decypher what does it mean.

Comments on the Quality of English Language

An Editorial control of the style and grammar appears to de desirable.

Author Response

(1) First of all the use of abbreviations should be explained at their first appearance in the manuscript (even in the Abstract!).

Response (1):

Thank you for your suggestion. We have rechecked the entire text and explained the abbreviations when they first appear.

(2) Secondly a mother language control of the English appears to be desirable, for example already the TITLE is formulated in a manner that one can/could not exactly decypher what does it mean.

Response (2):

Thank you for your suggestion. We have revised the title “Ganoderma lucidum immunomodulatory protein has the potential to improve the prognosis of breast cancer through the regulation of key prognosis‐related genes”.

Reviewer 2 Report

Comments and Suggestions for Authors

Could the author clarify why they chose a p-value criterion of <0.05 to find DEGs, and if stricter criteria were considered to reduce false positives?

What made author chose LASSO-Cox regression over other variable selection approaches, and what biases might this create in prognostic model?

What is the biological relevance of the six genes (SRGAP2, SLC35A2, FAM114A1, TP53I11, TMEM63C, and PIGR) utilized to create the predictive risk score and how they may affect patient outcomes?

The study revealed six prognostic genes for Luminal A breast cancer. Why were these genes (SRGAP2, SLC35A2, FAM114A1, TP53I11, TMEM63C, and PIGR) chosen for the prognostic model and how do they affect breast cancer progression?

FIP-glu controlled and reversed six genes, according to the investigators. Can author explain how FIP-glu precisely regulates these gene expressions in breast cancer cells?

The findings reveals immune system regulation is crucial to Luminal A breast cancer growth and prognosis. Could author explain how differentially expressed genes (DEGs) may affect immune-related circuits and how this knowledge could be used in immunotherapy?

The functional enrichment analysis identified immune-related GO keywords and KEGG pathways. How do these immune functions match Luminal A breast cancer features and offer treatment targets?

TMEM63C-overexpressing T-47D and MCF-7 cells showed anticancer effects of FIP-glu. Could author explain how FIP-glu counteracts TMEM63C's tumorigenic effects?

TP53I11 needs more research, according to the report. What aspects of TP53I11's role in breast cancer progression do authors think need additional study, and how would it improve understanding of this gene as a therapeutic target?

Comments on the Quality of English Language

The English could be improved.

Author Response

(1) Could the author clarify why they chose a p-value criterion of <0.05 to find DEGs, and if stricter criteria were considered to reduce false positives?

Response:

Thank you for your suggestion. We chose adjusted p < 0.05 to find DEGs.

(2) What made author chose LASSO-Cox regression over other variable selection approaches, and what biases might this create in prognostic model?

Response:

Thank you for your suggestion. LASSO-Cox regression is very commonly used in a large amount of literature and has been proved for the accuracy. For example, Liu et al. developed a hepatocellular carcinoma prognosis prediction model using LASSO-Cox regression to find fragmentomic features which exhibited exceptional ability [1]. Li et al. also used LASSO-Cox regression to develop a radiomics signature for biochemical recurrence following radical prostatectomy [2]. This model is a valuable complement for traditional analytic approaches and is a promising tool for future model development [3].

References:

  1. Liu,Y.; Peng, F.; Wang, S.; Jiao, H.; Zhou, K.; Guo, W.; Guo, S.; Dang, M.; Zhang, H.; Zhou,W.; Guo, X.; et al.Aberrant fragmentomic features of circulating cell-free mitochondrial DNA enable early detection and prognosis prediction of hepatocellular carcinoma. Clin Mol Hepatol, 2024.
  2. Li,T.; Xu, M.; Yang, S.; Wang, G.; Liu, Y.; Liu, K.; Zhao, K.; Su, X. Development and validation of [18 F]-PSMA-1007 PET-based radiomics model to predict biochemical recurrence-free survival following radical prostatectomy. Eur. J. Nucl. Med. Mol. 2024, 51, 2806-2818
  3. Ho, C.T.; Tan, E.C.; Su, C.W. Correspondence to editorial on "Conventional and machine learning-based risk scores for patients with early-stage hepatocellular carcinoma". Clin Mol Hepatol. 2024, 30, 1016-1018

(3) What is the biological relevance of the six genes (SRGAP2, SLC35A2, FAM114A1, TP53I11, TMEM63C, and PIGR) utilized to create the predictive risk score and how they may affect patient outcomes?

Response:

Thank you for your suggestion. The expression level of these genes correlated with survival probability of Luminal A breast cancer patients (p < 0.05). The Kaplan–Meier survival analysis of each gene based on its expression was constructed as follow.

(4) The study revealed six prognostic genes for Luminal A breast cancer. Why were these genes (SRGAP2, SLC35A2, FAM114A1, TP53I11, TMEM63C, and PIGR) chosen for the prognostic model and how do they affect breast cancer progression?

Response:

Thank you for your suggestion. The six genes were obtained by LASSO-Cox regression according to their expression levels and the relationship with the expression and the survival rate of breast cancer patients. We described the functions of each gene in the discussion. Some genes have been previously shown to participate in breast cancer progression. SRGAP2 was involved in breast cancer cell migration and the cancer process [1]. This gene serves as a linker to transmit the mechanical signals among cytoskeleton and membrane. It cooperates with cytoskeleton tension to participate in matrix-directed cell migration in breast cancer [1]. Sun et al. showed that SLC35A2 expression level was increased in breast cancer [2]. The high SLC35A2 expression was linked to increased immune infiltration in myeloid-derived suppressor cells, as well as immune checkpoints, ferroptosis-related genes, tumor mutational burde, and microsatellite instabilit. And knockdown of SLC35A2 could inhibit tumor growth in vivo. FAM114A1 has been identified as a risk gene in breast cancer. It is highly expressed in tumor tissues of breast [3]. But the mechanism by which it is associated with cancer has not yet been studied. TP53I11 can suppress epithelial-mesenchymal transition and metastasis in breast cancer cell. Xiao et al. reported that TP53I11 may suppress EMT and metastasis by reducing HIF1α protein levels in breast cancer cells [4]. In breast cancer, PIGR is shown to be downregulated in tumor tissues compared to normal tissue [5], and high PIGR expression is strongly associated with better cumulative survival compared to low PIGR level [6]. While the functions of TMEM63C in breast cancer have not been reported. These genes have the valuable potential to become new targets for treating breast cancer with further research.

References:

  1. Li, C.; Zheng, Z.; Wu, X.; Xie, Q.; Liu, P.; Hu, Y.; Chen, M.; Liu, L.; Zhao, W.; Chen, L.; et al. Stiff matrix induced srGAP2 tension gradients control migration direction in triple-negative breast cancer. Theranostics 2023, 13, 59-76.
  2. Sun, X.; Yuan, Z.; Zhang, L.; Ren, M.; Yang, J.; Xu, Y.; Hao, J. Comprehensive analysis of SLC35A2 in pan-cancer and validation of its role in breast cancer. J Inflamm Res 2023, 16, 3381-3398.
  3. Ren, X.; Cui, H.; Wu, J.; Zhou, R.; Wang, N.; Liu, D.; Xie, X.; Zhang, H.; Liu, D.; Ma, X.; et al. Identification of a combined apoptosis and hypoxia gene signature for predicting prognosis and immune infiltration in breast cancer. Cancer Med 2022, 11, 3886-3901.
  4. Xiao, T.; Xu, Z.; Zhang, H.; Geng, J.; Qiao, Y.; Liang, Y.; Yu, Y.; Dong, Q.; Suo, G. TP53I11 suppresses epithelial-mesenchymal transition and metastasis of breast cancer BMB Rep 2019, 52, 379-384.
  5. Bao, Y.L.; Wang, L.; Shi, L.; Yun, F.; Liu, X.; Chen, Y.X.; Chen, C.; Ren, Y.N.; Jia, Y.F. Transcriptome profiling revealed multiple genes and ECM-receptor interaction pathways that may be associated with breast cancer. Mol. Biol. Lett. 2019, 24.
  6. Amini, P.; Nassiri, S.; Malbon, A.; Markkanen, E. Differential stromal reprogramming in benign and malignant naturally occurring canine mammary tumours identifies disease-modulating stromal components. Rep. 2020, 10, 5506.

(5) FIP-glu controlled and reversed six genes, according to the investigators. Can author explain how FIP-glu precisely regulates these gene expressions in breast cancer cells?

Response:

Thank you for your suggestion. FIP-glu has excellent antitumor effects, but the mechanisms are complex and do not go far enough. Most studies have demonstrated its ability to suppress tumors by inducing certain forms of death [1-3], and the underlying mechanism requires in-depth study. The functions and regulation of these prognostic genes have not been reported well. And we first reported that all of these genes could be regulated by FIP-glu. The reviewer gives us a good suggestion and we will continue to do this work.

References:

  1. You, R.I.; Wu, W.S.; Cheng, C.C.; Wu, J.R.; Pan, S.M.; Chen, C.W.; Hu, C.T. Involvement of N-glycan in Multiple Receptor Tyrosine Kinases Targeted by Ling-Zhi-8 for Suppressing HCC413 Tumor Progression. Cancers (Basel). 2018, 1.
  2. Lin, T.Y.; H.Y. Hsu, Ling Zhi-8 reduces lung cancer mobility and metastasis through disruption of focal adhesion and induction of MDM2-mediated Slug degradation. Cancer Lett. 2016, 375, 340-348.
  3. Lin, T.Y.; Hua, W.J.; Yeh, H.; Tseng, A.J. Functional proteomic analysis reveals that fungal immunomodulatory protein reduced expressions of heat shock proteins correlates to apoptosis in lung cancer cells. Phytomedicine. 2021. 80, 153384.

(6) The findings reveal immune system regulation is crucial to Luminal A breast cancer growth and prognosis. Could author explain how differentially expressed genes (DEGs) may affect immune-related circuits and how this knowledge could be used in immunotherapy?

Response:

Thank you for your suggestion. According to the results of 2.4, we found that these differential genes may affect pathways such as T cell receptor signaling pathway, natural killer cell mediated cytotoxicity, and Th1 and Th2 cell differentiation. Targeting these signaling pathways are classic approach in tumor immunotherapy. In particular, the rapidly expanding capacity to definitively link intratumoural phenotypes with the antigen specificity of T cells provided by T cell receptors has now made it possible to focus on investigating the properties of T cells with tumour-specific reactivity [1]. For further research, we may design experiments related to T cells to study the role of FIP-glu in tumor immunotherapy.

References:

  1. Oliveira, G.; Wu, C.J. Dynamics and specificities of T cells in cancer immunotherapy. Nat Rev Cancer. 2023, 23, 295-316.

(7) The functional enrichment analysis identified immune-related GO keywords and KEGG pathways. How do these immune functions match Luminal A breast cancer features and offer treatment targets?

Response:

Thank you for your suggestion. Tumors are commonly classified as luminal A (ER+, PR−/PR+, HER2−), luminal B (ER+, PR−/PR+, HER2+), HER2 (ER−, PR−, HER2+), and TNBC (triple-negative breast cancer) (ER−, PR−, HER2−) based on receptor status[1]. Luminal A breast cancer is the largest subtype of breast cancer. Unlike other subtypes, most luminal A breast cancers are immune deserts; the underlying mechanisms are poorly understood. Ishikawa et al. reported that low levels of MIP-1b in luminal A breast cancers results in low induction of TILs [2]. And Wu et al found that increased frequencies of CD8 + PD1+, CD8 + TIM3 + and CD4 + Foxp3 + T cells might inhibit the immune microenvironment of axillary metastatic lymph nodes in luminal A breast cancer patients and subsequently promote lymph node metastasis [3]. There are no other previous reports of immune function regulation in Luminal A breast cancer. The correlation pathways we obtained through GO keywords and KEGG pathways are only high-scoring signaling pathways obtained through correlation, and further experimental exploration is needed to study and prove the specific mechanism.

References:

  1. Cheang, M.C.; Martin, M.; Nielsen, T.O.; Prat, A.; Voduc, D.; Rodriguez-Lescure, A.; Ruiz, A.; Chia, S.; Shepherd, L.; Ruiz-Borrego, M.; et al. Defining breast cancer intrinsic subtypes by quantitative receptor expression. Oncologist. 2015, 20, 474-82.
  2. Wu, M.; Wang, S.; Yuan, K.; Xiong, B.; Li, Y.; Lyu, S. Alteration of the immune microenvironment in the axillary metastatic lymph nodes of luminal A breast cancer patients. World J. Surg. Oncol. 2024, 22, 172.
  3. Ishikawa, E.; Watanabe, T.; Kihara, T.; Kuroiwa, M.; Komatsu, M.; Urano, S.; Nagahashi, M.; Hirota, S.; Miyoshi, Y. The cytokine profile correlates with less tumor-infiltrating lymphocytes in luminal A breast cancer. Breast Cancer Res. Treat. 2024.

(8) TMEM63C-overexpressing T-47D and MCF-7 cells showed anticancer effects of FIP-glu. Could author explain how FIP-glu counteracts TMEM63C's tumorigenic effects?

Response:

Thank you for your suggestion. The functions of TMEM63C in breast cancer have not been reported. Based on the model, we found that the expression of TMEM63C could affect prognosis. Based on big data, it is found that TMEM63C expression is upregulated. Thus, it is speculated that TMEM63C might be a cancer-promoting gene. Then, we over-expressed TMEM63C in T-47D and MCF-7 cells to confirm the speculation. Furthermore, we also found that FIP-glu counteracted TMEM63C's tumorigenic effects and the mechanism is not clear. This is our next work plan.

(9) TP53I11 needs more research, according to the report. What aspects of TP53I11's role in breast cancer progression do authors think need additional study, and how would it improve understanding of this gene as a therapeutic target?

Response:

Thank you for your suggestion. Pan-cancer analysis of TP53I11 showed that it is associated with low survival rates for breast cancer. In breast cancer, TP53I11 knockdown inhibits EMT and metastasis [1], but no further studies have been conducted. It will be an interested topic of the research for how TP53I11 regulates the proliferation and metastasis of breast cancer cells, whether it is a resistance gene, and whether it is druggable. These will be considered in our further study.

References:

  1. Xiao, T.; Xu, Z.; Zhang, H.; Geng, J.; Qiao, Y.; Liang, Y.; Yu, Y.; Dong Q.; Suo, G. TP53I11 suppresses epithelial-mesenchymal transition and metastasis of breast cancer cells. BMB Rep. 2019, 52, 379-38.

Round 2

Reviewer 2 Report

Comments and Suggestions for Authors

Authors have modified the article as per reviewer comments. Accept in present form.

Author Response

Comments 1: Authors have modified the article as per reviewer comments. Accept in present form.

Response 1: Thank you for recognizing our work and we will continue to work hard to publish better research.